# Effect of Salinity on Leaf Functional Traits and Chloroplast Lipids Composition in Two C_3_ and C_4_ Chenopodiaceae Halophytes

**DOI:** 10.3390/plants11192461

**Published:** 2022-09-21

**Authors:** Olga Rozentsvet, Elena Shuyskaya, Elena Bogdanova, Viktor Nesterov, Larisa Ivanova

**Affiliations:** 1Samara Federal Research Scientific Center Russian Academy of Sciences, Institute of Ecology of Volga River Basin, Russian Academy of Sciences, 445003 Togliatti, Russia; 2K. A. Timiryazev Institute of Plant Physiology, Russian Academy of Sciences, 127276 Moscow, Russia; 3The Institute of Environmental and Agricultural Biology (X-BIO), Tyumen State University, 625003 Tyumen, Russia

**Keywords:** C_3_, C_4_, cell morphology, chloroplast, fatty acids, halophytes, leaf anatomy, lipids membrane

## Abstract

Salt stress is one of the most common abiotic kinds of stress. Understanding the key mechanisms of salt tolerance in plants involves the study of halophytes. The effect of salinity was studied in two halophytic annuals of Chenopodiaceae *Salicornia perennans* Willd. and *Climacoptera crassa* (Bied.) Botsch. These species are plants with C_3_ and C_4_-metabolism, respectively. We performed a comprehensive analysis of the photosynthetic apparatus of these halophyte species at different levels of integration. The C_3_ species *S*. *perennans* showed larger variation in leaf functional traits—both at the level of cell morphology and membrane system (chloroplast envelope and thylakoid). *S*. *perennans* also had larger photosynthetic cells, by 10–15 times, and more effective mechanisms of osmoregulation and protecting cells against the toxic effect of Na^+^. Salinity caused changes in photosynthetic tissues of *C*. *crassa* such as an increase of the mesophyll cell surface, the expansion of the interface area between mesophyll and bundle sheath cells, and an increase of the volume of the latter. These functional changes compensated for scarce CO_2_ supply when salinity increased. Overall, we concluded that these C_3_ and C_4_ Chenopodiaceae species demonstrated different responses to salinity, both at the cellular and subcellular levels.

## 1. Introduction

Salt stress is one of the most common abiotic stressors. The adaptation of halophytes to salinity is associated with a large number of adaptive mechanisms and types of ecological strategies [1,2]. Euhalophyte cells are effective in moving Na^+^ out of their cytoplasm and maintaining the required K^+^ levels [3,4,5]. Cellular Na^+^/K^+^ homeostasis is tightly associated with the production of reactive oxygen species (ROS) [6,7,8], causing oxidative stress and subsequent damage to cellular biomolecules [9]. Halophytes possess well-regulated antioxidant systems for quenching toxic ROS produced under saline conditions [10,11,12]. In particular, the amino acid proline stabilizes proteins and membranes through the formation of a hydrophilic envelope, which in turn prevents the inactivation of proteins by hydroxyl radicals and singlet oxygen [13,14].

Among key mechanisms, the pathway of photosynthesis plays an important adaptative role [15,16]. Among the dicot families, the Chenopodiaceae has the largest number of C_4_ species, and consequently the greatest diversity in leaf anatomy [17]. Many Chenopodiaceae species grow on highly saline soils [18]. Another method of plant adaptation to salinity is succulence. Succulent halophytes have morphological and anatomical features that ensure their effective response to salt. In halophytes, the main function of a succulence is to achieve a better water-use efficiency and maintain turgor of their succulent organs [10,19]. The succulence of euhalophyte leaves is closely associated with the accumulation of ions in their vacuoles [20]. In addition, succulence is usually accompanied by an increase in cell size. This makes it possible to increase the chloroplast number when under salt stress [21].

Analysis of transverse sections of stem fragments of *Salicornia* (*Amarantaceae*) using fluorescence microscopy revealed that cells in the palisade parenchyma had a high number of chloroplasts and specific cells with expandable vacuoles [22].

The main difference between C_4_ and C_3_ plants is in the spatial separation of light and dark phases in photosynthesis, supported by the specific Kranz-anatomy of the leaf [23]. The anatomical structure of the leaves of C_3_ plants is usually represented by cells of the palisade and spongy parenchyma, which do not differ in metabolism. Dimorphic and functionally different phototrophic tissues—mesophyll and bundle sheath—are considered necessary for the functioning of C_4_ photosynthesis [24,25]. The two types of photosynthesis are associated with specific anatomical features of the photosynthetic organs and the involvement of different carboxylation enzymes. Regulation of photosynthetic function occurs by changing the quantitative parameters of photosynthetic tissues [26,27] and the content of the main pigments of photosynthesis [28]. Their changes are associated, both with the number of chloroplasts per unit leaf area, and with the functional characteristics of a single chloroplast [29,30].

The structure of photosynthetic membranes plays an important role in the resistance of halophytes to salinity [7,31,32]. Lipids provide mobility and dynamic properties of chloroplast membranes [33]. Pigments are integrated into thylakoid membranes and involved in sites of light reactions of photosynthesis [34]. The structural components of thylakoid membranes are the lipids monogalactosyldiacylglycerol (MGDG) and digalactosyldiacylglycerol (DGDG), sulfoquinovosyldiacylglycerol (SQDG), and phosphatidylglycerol (PG) [35]. The functional state of thylakoid membranes is determined by the composition of lipids and their fatty acids (FA).

In this work, we hypothesized that plants with different types of photosynthetic metabolism can differ significantly in mechanisms and the direction of changes in the structure and function of leaves, cells, chloroplasts, and photosynthetic membranes with salinity.

## 2. Results

### 2.1. Region and Conditions of Study

The region of study was in the northern part of the Caspian Lowland (Volgograd region, Russian Federation, 49°07′ N. latitude, 46°50′ E. longitude). Two halophyte species *Salicornia perennans* Willd. (C_3_-type of photosynthesis) (further Sal-1, Sal-2) and *Climacoptera crassa* (Bied.) Botsch. (C_4_-NAD-type) (Cl-1, Cl-2) were selected for the study. The experimental material was taken from specially selected ecotopes with different soil salinity and soil moisture (Table 1). Sal-1 and Sal-2 were located near the water edge of Lake Elton dominated by *S*. *perennans*. The ecotopes Cl-1 and Cl-2 had a low degree of humidity and salinity soil, typical for the species *C*. *crassa*.

### 2.2. Contents of Na^+^ and K^+^ in the Leaves

The level of Na^+^ was found to be equal in the tissues of both species growing in either less saline or more saline ecotopes (Figure 1A). The concentration of K^+^ in the tissues of *S. perennans* in the more saline ecotope, Sal-2, was 30% higher than in the Sal-1 ecotope (Figure 1B). In contrast to *S. perennans*, the accumulation of K^+^ ions in the leaves of *C. crassa* did not differ between ecotopes. As a result, the Na^+^/K^+^ ratio in the tissues of *C. crassa* was higher than that in *S. perennans* (Figure 1C).

### 2.3. Functional Traits of the Leaves

The water content in *S. perennans* was approximately 10% higher than that in *C. crassa* (Figure 2A). Increased soil salinity had no effect on the water content in the leaves of both plants. However, in response to changes in salinity, the total amount of chlorophylls in *C. crassa* increased by 1.6 times and did not change in *S. perennans* (Figure 2B). In *S. perennans* (C_3_), the level of lipid peroxidation (LPO) products, measured by malondialdehyde (MDA) concentration in dry weight, was lower than in *C. crassa* (C_4_) (Figure 2C). Salinity caused a reduction in MDA level in the *S. perennans*, and a 1.6-fold increase of MDA level in the *C. crassa*. The same tendency was observed for proline: in *S. perennans*, it decreased by 3.5 times, while in *C. crassa*, it increased by 1.3 times (Figure 2D).

### 2.4. Leaf Morphological Traits

Photosynthetic organs of the C_3_ species *S. perennans* had an undifferentiated chlorenchyma. The leaf anatomy of the C_4_ species *C. crassa* was characterized by its Kranz-anatomy, i.e., clear differentiation of photosynthetic tissues into mesophyll (M) and bundle sheath (BS). The BS had larger cell size, as compared to the M cells (Figure 3A). However, the total number of photosynthetic cells (bundle sheath + mesophyll) per unit of leaf area was similar in *C. crassa* and *S. perennans* growing in low-saline ecotopes (Figure 3B). The volume of mesophyll cell increased 4-fold in *S. perennans* and 2-fold in *C. crassa* in high-saline ecotopes. In *C. crassa*, a 1.8-fold increase in the volume of bundle sheath cells (Vcell (BS)) was also observed (Figure 3B). Changes in the cell number per leaf area were inversely proportional to the changes in their volume: the larger the cells, the lower their number per unit of leaf area in both species.

The studied species also differed by the number and size of chloroplasts in photosynthetic cells. In the mesophyll cells of the C_3_ species, the chloroplast volume and number were 1.4 and 2.7 times higher than those in the mesophyll cells of C_4_ species (Figure 4A,B). At high salinity, the chloroplast number per cell in *S. perennans* increased 2-fold (Figure 4B). The number of chloroplasts per unit of leaf area was higher in *S*. *perennans*, and was not affected by soil salinity (Figure 4C). In the plants of *C. crassa*, the number of chloroplasts per leaf area increased with salinity. Cell volume per one chloroplast increased 1.7-fold in the C_3_ species (Figure 4D) and 2-fold in the mesophyll cells of the C_4_ species.

The changes in the number of cells and chloroplasts affected the integral parameters of the mesophyll. The leaves of the C_3_ *S*. *perennans* at high salinity became 25% thicker and the number of cells decreased by half but preserved the total surface area of photosynthetic cells per unit of leaf area (Ames/A) (Table 2). In the C_4_ species, these characteristics increased 2-fold in more saline ecotopes. In addition, the ratio of mesophyll-to-bundle sheath cells in the C_4_ *C*. *crassa* diminished by 25% (Table 2).

### 2.5. Lipids of Chloroplast Membranes

The total content of membrane lipids in the chloroplasts of the examined plants varied in the range of 5–10 mg·g^−1^ dry weight (DW) (Table 3). In addition to MGDG, DGDG, SQDG, and PG, the extracts of chloroplast lipids contained PC, phosphatidylethanolamine (PE), phosphatidylinositol (PI), and phosphatidic acid (PA), as well as small amounts of lysophospholipids (≤0.5% of total lipids), and sterols (2–6% of total lipids) (Table 3). The total content of membrane lipids in the chloroplasts of *S. perennans* plants growing on Sal-2 increased compared with Sal-1—as did the volume of their cells and the number of chloroplasts per cell (Figure 3). In contrast, *C. crassa* plants from highly saline ecotopes did not show any increase in the total content of chloroplast membrane lipids per DW.

The *S. perennans* plants growing on highly saline soils showed a notable drop in the proportion of MGDG, PG, and SQDG in their thylakoid membranes, whereas the proportion of PC in their chloroplast membranes increased. The ratio of the two neutral lipids (MGDG/DGDG), as well as the ratio of charged-to-neutral lipids (MGDG + DGDG/SQDG + PG) in those plants decreased by the factors of 1.4 and 1.7, respectively. The lipid composition of chloroplast membrane of the C_4_ species remained stable. There were also changes in the PC/PE ratio: in *S. perennans*, it increased almost 3-fold; in *C. crassa*, it decreased by a factor of 1.5.

The content of FA with the chain length of 16 and 18 carbon atoms was above 90% of their sum (Table 4). The C_4_ species, *C. crassa*, differed from the C_3_ species, *S. perennans*, by a much higher percentage of oleic (C18:1n9c) acid (~5 times as high). At the same time, the levels of linoleic (C18:2n6c) and linolenic (18:3n3) acids were higher in *S. perennans*. For both species, the content of palmitic FA (16:0) increased by 1.2-fold and the content of 18:3n3 decreased in highly saline ecotopes compared to less saline ones.

## 3. Discussion

Halophytes of the Elton region occupy strictly defined ecotopes with a certain salinity and soil moisture [18,36]. For example, halophytic annual *S*. *perennans* grows in the most mineralized and wettest ecotopes. In comparison, less saline and humid conditions are typical for *C*. *crassa*. It is known that different soil salinity causes specific anatomical, morphological, and physiological traits, including different types of photosynthesis [37,38]. Both species use the salt-accumulating strategy [17]. Our study revealed no significant differences in the content of Na^+^ in leaves between species, despite tenfold differences in soil salinity between paired ecotopes Sal-1/Sal-2 and Cl-1/Cl-2 (Figure 1). In nature, the habitats of *C*. *crassa* plants must intensively accumulate, by 9–14-fold, more Na^+^ to achieve the required Na+ level in the leaves from less saline soil as compared to c *S*. *perennans* (Table 1, Figure 1). The Na^+^plant/Na^+^soil ratio for *C*. *crassa* was also 9–15-fold higher than that for *S*. *perennans* (Table 5). The differences in the Na^+^plant/Na^+^soil ratios in plants from less and more saline ecotopes reflect differences in the Na^+^ content in the soil of these ecotopes.

The K^+^ content in *C. crassa* leaves was similar in both ecotopes, while K^+^ content in *S. perennans* leaves was different (Figure 1). Under salinity, maintaining the optimal K^+^/Na^+^ ratio in the cytosol is critical for the normal functioning of the cytoplasm [21,39]. It has been suggested that in chloroplasts of halophytes, K^+^ can be replaced by Na^+^ without any harmful effect on photosynthesis [21,40]. However, it has been shown that halophytes retain at least 20% more K^+^ in chloroplasts compared to the cytoplasm [41]; and K^+^ cannot be substituted for Na^+^ in chlorophyll synthesis and Rubisco synthesis [22,42]. To maintain the required level of K^+^ in leaves, *C. crassa* also accumulates K^+^ more intensively, due to its 2–11-fold lower amount in the soil, compared to *S. perennans* (Table 1 and Table 5). The selectivity for K^+^ over Na^+^ (net S_K_:_Na_) in *S. perennans* is higher than in *C. crassa* (Table 5), which is probably due to the high Na^+^ content in the soil of Sal-1 and Sal-2 ecotopes, and the need to maintain a certain Na^+^/K^+^ balance in leaves (Table 1). The constant amounts of ions in leaves of these halophytes under different contents of Na^+^, K^+^, and water in the substrate indicates the presence of mechanisms for maintaining a strictly defined Na^+^/K^+^ balance in both species.

*S. perennans* plants demonstrated more active accumulation of Na^+^ in a less saline ecotope, Sal-1, which may lead to a more rapid flow of ions into tissues and their accumulation in the apoplast, also as a consequence of a smaller volume of cells (Figure 3A). It is quite possible to assume that in the Sal-2 ecotope, the main part of Na^+^ enters the cell, rather than the apoplast, and thus, takes part in the regulation of the cell wall and cell size [43]. There is a decrease in the number of cells per unit leaf area under more saline conditions (Sal-2), which allows *S. perennans* plants to maintain the total surface area of cells and chloroplasts per unit leaf area at the same level as in plants growing on less saline soil Sal-1 (Table 2). As a result, the total intra leaf assimilation surface in the C_3_ species did not depend on the degree of soil salinity.

In *C. crassa*, increasing soil salinity caused the increase of total surface of mesophyll (Ames/A) and chloroplasts per leaf area (Achl/A) (Table 2). The interface area between the mesophyll and bundle sheath cells also rose due to an increased volume of sheath cells (Figure 3A). A decreased ratio of mesophyll-to-bundle sheath cells, along with an increase in their volume ratio, mirrors a general shift in the ratio of mesophyll tissue-to-bundle sheath. However, the volume and number of mesophyll cells, as well as the number of chloroplasts per unit leaf area in those species, did not change. Our results show that plants with different types of photosynthesis use different strategies of adaptation to salinization at the level of photosynthetic tissues.

There is a direct causal and evolute connection between studied traits of salt tolerance and a type of photosynthetic metabolism. An increase in mesophyll cell volume and succulence are shown as common features of many C_3_ and C_4_ species tolerant to high salinity [44,45]. Our results demonstrated a 4-fold increase in cell size in C_3_ *S*. *perennans* and a 1.8-fold increase in the volume of bundle sheath cells in C_4_ *C*. *crassa* under salinity. At the same time, C_3_ plants have a larger variation in cell size response to salinity than C_4_ plants. The occurrence of C_4_ syndrome is another mechanism of salt tolerance different and independent from succulence. It is shown that the abundance of C_4_ plants correlates with salinity [37,46]. Therefore, differences in cell volume and in the limits of changes in cell sizes between C_3_ and C_4_ taxonomically relative annuals demonstrate the different evolute mechanisms of response to salinity in C_3_ and C_4_ plants.

We believe that the studied functional traits have different relevance to plant salt adaptation. Some traits are applied to C_3_ and C_4_ plants in general. High mesophyll volume, as well as larger variation in quantitative leaf traits in C_3_, especially in mesophyll cell volume, in comparison to C_4_, belong to the general features of C_3_ halophytes [38]. Some other traits are characteristic to definite plant functional types—C_3_ or C_4_ (NAD-Me)-plants. It is known that the abundance of C_4_ plants correlates with salinity [38,47]. Other types of C_3_ halophytes, such as non-succulents and other types of C_4_-plants, will have different traits. Within C_3_-halophytes, such traits specific for succulence and non-succulence are leaf mass per area, cell volume, cell number, total area of mesophyll cell surface per leaf area [48]. The ratio in quantitative traits between mesophyll and bundle sheath (M/BS)—cell number M/BS, cell volume M/BS, chloroplast number M/BS, chloroplast volume M/BS, and the variation in interface area between mesophyll and bundle sheath cells—are specific for different C_4_-photosynthesis subtypes [49,50]. Some studied functional traits are specific for these studied species—leaf thickness and chloroplast number per cell are shown to be strongly species-specific in plants of sub-arid conditions [30].

Proline accumulation is one of the most notable changes in plant metabolism in response to salt stress. In our study, there is a clear relation between the proline content and the lipid oxidation level (MDA) (Figure 2C,D). The relationship and mutual regulation of ROS and proline metabolisms are well known. ROS, as signaling molecules, can regulate proline biosynthesis. Proline in the studied halophytes plays a more significant role in the salt tolerance of the C_3_ species as compared to the C_4_ species.

The chlorophyll content in *S. perennans* was the same at both study sites (Figure 2B), while *C. crassa* showed an increased chlorophyll level in more saline conditions (Figure 2C,D). Increase in chlorophyll content was caused by leaf thickening along with enhancement in chloroplast number per leaf area. Our results indicate that structural changes in leaf and increased chlorophyll content in C_4_ plants may serve as a response to increased soil salinity.

Chloroplasts of *S. perennans* in the Sal-2 ecotope contain more membrane lipids compared to those in less saline Sal-1 ecotopes. Reduced MGDG/DGDG and MGDG + DGDG/SQDG + PG ratios observed in the chloroplasts of *S. perennans* growing in the Sal-*2* ecotope, indicate changes in the architectonics of thylakoid membranes, thylakoid profile, and ratio of granal and agranal thylakoids [7,34,51], whereas the increased PC/PE ratio indicates an alteration in the asymmetry of their thylakoid membranes [32], as a result of which the degree of membrane invagination increases, as well as its permeability decreases due to a decrease in monolayer sites. PC and PE, as well as MGDG and DGDG, differ in their ability to form a membrane bilayer. PC and DGDG are a bilayer-forming lipid, and PE and MGDG form monolayers. Such a change in the asymmetry of thylakoid membranes is usually accompanied by an increase in the degree of their curvature and a decrease in their permeability [52]. In contrast to the C_3_ species *S. perennans*, C_4_ *C. crassa* appears to have less opportunities for rearrangement, confirmed by stability of their lipid composition. The FA composition of chloroplast lipids also changes: the content of the saturated acid, 16:0, grows, and the content of the unsaturated acid, 18:3n3, decreases. Similar changes in the ratio of the main unsaturated FA in the photosynthetic organs of halophytes were reported earlier [53]. In contrast to the C_3_ species, the membranes of the C_4_ species of *C. crassa* likely have fewer opportunities for rearrangement of photosynthetic membranes, confirmed by the stability of their lipid and FA composition.

## 4. Materials and Methods

### 4.1. Plants Material

Plant material and soil samples were collected in the first half of June 2018. The study was conducted under the conditions of high insolation (1000–2000 PPFD µmol m^−2^ s^−1^) and temperature at day/night of 30–35/25–30 °C. The middle part of leaves/ succulent stems of 15–20 plants collected within the same phytocenosis were used for biochemical analyses. Three independent biological samples for each analysis type (2–4 g) of fresh weight (FW) were made from overall leaf biomass and frozen in liquid nitrogen. Soil samples were taken at a depth of 15–20 m^−2^ to determine the mineral residue and moisture content [54].

### 4.2. The Ions and Water Contents in Soil and Leaves

For the analysis of ions and humidity of soils, samples were collected at a depth of 15–20 cm. The content of Na^+^ and K^+^ were determined by atomic absorption spectrometry (MGA 915, Lumex, St. Petersburg, Russia). The content of water was calculated after drying the soil samples to a constant weight at 60 °C, and the results were expressed as % of fresh weight (FW). The contents of ions in the leaf tissues were determined in mineralized and milled samples by the method of ICP-OES (inductively coupled plasma optical emission spectroscopy) using an atomic absorption spectrometer (MGA 915, Lumex, St. Petersburg, Russia). Net K:Na selectivity (S_K:Na_) was calculated as the ratio of K^+^/Na^+^ in the plant divided by K^+^/Na^+^ in the medium [41]. We determined the FW of plants. Plants were then dried at 60 °C for 2 days and weighed to measure their DW. Water content in plant leaves was calculated according to the formula: W = (FW–DW)/FW × 100%.

### 4.3. Pigment, Malondialdehyde and Proline Content

The chlorophylls *a* and *b* were extracted from 0.2–0.5 g of fresh leaves with 80% acetone. The content of pigments was determined at 662 and 645 nm using a PE-3000 UV spectrophotometer (PromEcoLab, Shanghai, China). The contents of chlorophylls *a*, *b* (g^−3^ g^−1^ DW) were calculated according to [55].

The lipid peroxidation in plant leaves was estimated from the content of MDA [9]. The MDA reacts with thiobarbituric acid (TBA), forming a pink chromogen thiobarbituric acid reactive substance (TBARS). A weighed sample of fresh leaves (0.5 g) was homogenized in 10 mL of isolation medium 0.1 M Tris-HCI (pH 7.6). To the homogenate (3 mL), 0.5% TBA in 20% trichloroacetic acid (2 mL) was added. The mixture was boiled in a water bath for 30 min. The content of TBARS was determined spectrophotometrically (PE-3000 UV, PromEkoLab, Saint Petersburg, Russia) at λ = 532 nm. The accumulation of TBARS was calculated by the formula *C* = D/(ɛ × *l*), where *C* is the concentration of TBARS, *D* is the optical density of the solution, is the molar extinction coefficient (0.156 × 10^6^ L cm^−1^ mol^−1^), and *l* is the thickness of the cuvette. The results were expressed in μmol·g^−1^ DW.

Free proline was determined according to [56] with modifications. Dry shoot or leaf samples (0.2 g) were homogenized in 2 mL of boiling distilled water, heated at 100 °C for 10 min in a water bath and then the homogenates were centrifuged (5 min, 14,000× *g*). The 1 mL of homogenate was reacted with 1 mL acidic ninhydrin (ninhydrin 1% (*w*/*v*) in acetic acid 60% (*v*/*v*), ethanol 20% (*v*/*v*)) and 1 mL glacial acetic acid in a tube for 1 h at 100 °C in a water bath, and the reaction terminated in an ice bath. The mixtures were read at 520 nm using a Genesis 10 UV Scanning spectrophotometer (Thermo Scientific, Waltham, MA, USA). Proline concentrations were determined using a calibration curve and expressed as μmol g^−1^ DW. 

### 4.4. Leaf and Mesophyll Traits

The leaves of *S. perennans* are fused with the internode of the stem, whereas *C. crassa* has thick succulent leaves. Correspondingly, the leaf traits of *S. perennans* were measured as the shoot projection area; in case of *C. crassa*, it was the leaf projection area. The cell and chloroplast number were counted on shoot /leaf fragments fixed in 3.5% glutaraldehyde solution in phosphate buffer (pH = 7.4). The number of cells per unit of leaf area (Ncell) was measured in a suspension obtained after the maceration of leaf samples in 20% KOH. The number of mesophyll and bundle sheath cells was counted in a Goryaev hemocytometer at a magnification of ×200 under a Zeiss Axiostar light microscope (Carl Zeiss, Oberkochen, Germany) in a cell suspension obtained after the maceration of leaf samples in 24 replicates [57,58]. The cell size and the number of chloroplasts per cell (Chl) were estimated in a cell suspension obtained after the maceration of leaf discs in 1 N HCl at 50 °C for 10 min in 30 replicates for both species. The cell volume (Vcell) was calculated by the projection method, based on average values of the cell projection area and perimeter, as well as coefficients dependent on the shape of cells in 30 replicates per sample [29]. The two-dimensional shape factor of cells (K_2D_) was calculated as a ratio of the squared projection perimeter to cell projection area [29]. Chloroplast sizes were measured on leaf cross sections using the Zeiss Axiostar light microscope (Carl Zeiss, Oberkochen, Germany) and a Simagis Mesoplant analyzer (SIAMS LLC, Ekaterinburg, Russia) using a projection method in 30 replicates per sample [58]. The chloroplast number per unit leaf area (Nchl) was calculated by multiplying the number of chloroplasts per cell (Chl) by the number of cells per unit of leaf area (Ncell) in 30 replicates per sample. The mesophyll area per unit leaf area (Ames/A) was calculated by multiplying the number of cells per leaf area (Ncell) by the cell surface area. The chloroplast area per unit leaf area (Achl/A) was calculated by multiplying the number of chloroplasts per leaf area (Ncell) by the chloroplast surface area.

### 4.5. Isolation of Chloroplasts

Chloroplasts were isolated from the plant tissues by differential centrifugation after homogenizing the tissues in a medium containing 0.5 M sucrose, 5 mM EDTA, 5 μM β-mercaptoethanol, and 50 mM Tris-HCl (pH = 7.8). The homogenate was centrifuged for 10 min at 500× *g* (chloroplast pellet). The pellets were resuspended in a medium containing 0.5 M sucrose and 5 mM Tris-HCl (pH = 7.2) [59]. The purity of chloroplast fractions was monitored using an Axio observer Z1 inverted biological microscope (Carl Zeiss, Oberkochen, Germany).

### 4.6. Lipid Extraction and Analysis

Chloroplast lipids were extracted from the suspension of the organelles with a chloroform/methanol mixture (1:2, *v*/*v*) and separated by thin layer chromatography (TLC) [60]. The lipids were quantified densitometrically using the Denscan-04 program (Lenchrome, St. Petersburg, Russia), developed specifically for processing chromatograms on TLC plates. In some cases, lipids were also quantified by their specific color reactions (specific spray-reagents were used to identify polar lipids: molybdenum blue and malachite green for phosphorus lipids; Dragendorff’s reagent for cholin containing lipids; and a 0.2%-solution of ninhydrin in acetone for amine lipids [60] using a PromEcoLab PE-3000 UV spectrophotometer. The densitograms were analyzed in the mode of parabolic approximation, using the calibration curves obtained with MGDG and PC (Sigma, Roedermark, Germany) [60].

FA were analyzed as methyl esters (FAME) using a Crystal 5000.1 gas chromatograph (Chromatek, Yoshkar-Ola, Russia). The analysis was performed in the isothermal mode on an Rtx T-2330 capillary column (*l* = 105 m, Ø = 0.25 mm; Restek, Bellefonte, PA, USA). The temperature of the column was 180 °C; the temperature of the evaporator and detector, 260 °C. The flow rate of the carrier gas (helium) was 2 mL·min^−1^. FAME were identified by comparing their retention times with FA standards (Supelco 37, Supelco, Bellefonte, PA, USA), and quantification was performed with heptadecanoate as the internal standard. This standard contains methyl esters of FA ranging from C4 to C24, including key monounsaturated and polyunsaturated FA.

### 4.7. Statistical Analyses

Two halophyte species from specially selected ecotopes with different soil salinity and soil moisture were studied. To test for differences between categories following two-way ANOVAs, Tukey post hoc tests were used. Differences were significant at *p* ≤ 0.05. In figures and tables, means ± SE are presented. Different letters are used in figures and tables to indicate significant differences. All statistical analyses were carried out in Statistica 6.0 (StatSoft Inc., Tulsa, OK, USA) and Statgraphics Centurion XVI (StatPoint Technologies, Warrenton, VA, USA).

## 5. Conclusions

The C_3_ species *S*. *perennans* is characterized by variable traits of the photosynthetic apparatus—both at the level of cell morphology and the level of chloroplast membrane systems (envelope and thylakoid membranes). As compared to the C_4_ species *C*. *crassa*, it also has more effective osmoregulatory and protecting mechanisms against the toxic effect of Na^+^. As a result, *S*. *perennans* withstands a high level of soil salinity subject to high soil humidity. The C_4_ species *C*. *crassa* grows on less saline and more dry soils, i.e., under the conditions of milder ionic, but more severe, osmotic stress. Adaptation of photosynthetic apparatus in *C*. *crassa* to salinity is aimed at the expansion of the intraleaf CO_2_-diffusion surface, expansion of the interface area between mesophyll and bundle sheath cells due to the increasing the volume of the latter. These adaptations should compensate for scarce CO_2_ supply when water is in deficit. Overall, the data obtained allow us to conclude that the two Chenopodiaceae species with C_3_ and C_4_ types of photosynthetic metabolism developed different mechanisms of adaptation to both at the cellular and subcellular levels.

Notably, the plants grew in ecotopes with a huge difference in soil salinity. Thus, different adaptation strategies can be caused, not only by C_3_ C_4_ photosynthesis, but by salinity level as well. It is likely that further experiments on the effects of the same salinity levels in controlled experiments may clarify this limitation.

## Figures and Tables

**Figure 1 plants-11-02461-f001:**
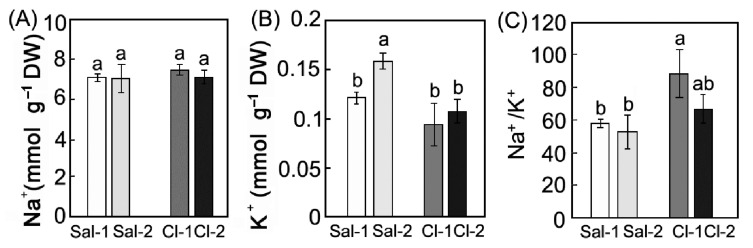
Content of Na^+^ (**A**), K^+^ (**B**), Na^+^ / K^+^ ratios (**C**) in the aboveground mass of plants with different types of photosynthesis: C_3_ *S. perennans* (Sal-1, Sal-2) and C_4_ *C. crassa* (Cl-1, Cl-2). Data represent means ± SE of *n* = 3 for each ecotopes. Different Latin letters indicate significant differences (two-way ANOVAs with Tukey post hoc tests, *p* < 0.05).

**Figure 2 plants-11-02461-f002:**
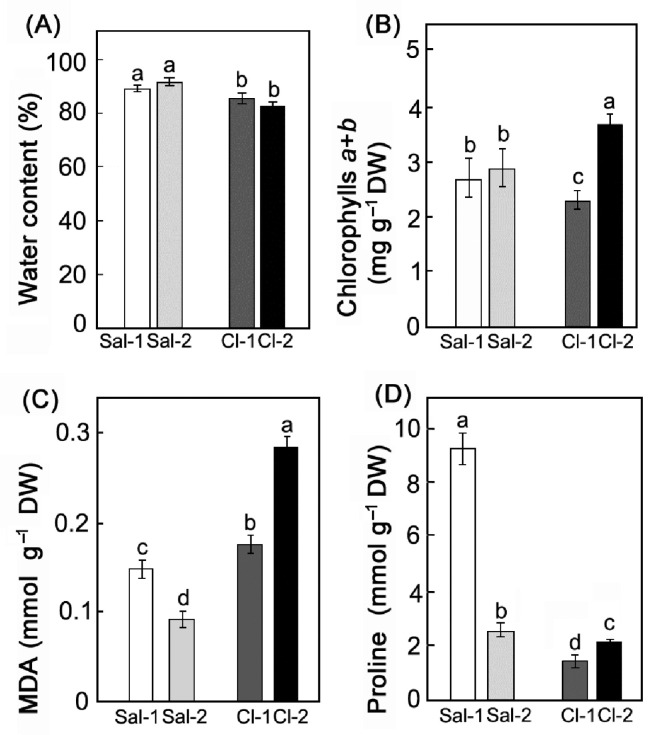
Effect of salinity on the content of water (**A**), chlorophylls *a* + *b* (**B**), the amount of MDA (**C**) and the content of proline (**D**) in plants with different types of photosynthesis: C_3_ *S. perennans* (Sal-1, Sal-2) and C_4_ *C. crassa* (Cl-1, Cl-2). Data represent means ± SE of *n* = 3 independent samples from the combined biomass of 2–3 leaves from 10–15 plants each species. Different Latin letters indicate significant differences (two-way ANOVAs with Tukey post hoc tests, *p* < 0.05).

**Figure 3 plants-11-02461-f003:**
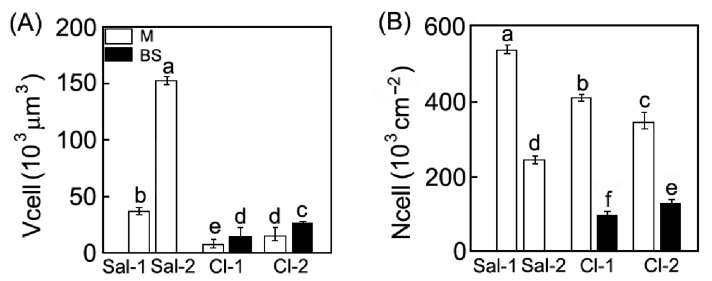
Effect of salinity on the cell volume (**A**) (Vcel) and cell number per unit leaf area (**B**) (Ncell) in *S. perennans* (Sal-1, Sal-2) and *C. crassa* (Cl-1, Cl-2). Data represent means ± SE of *n* = 30 for (**A**) and means ± SE of *n* = 24 for (**B**) from cell suspension obtained after the maceration of leaf samples for a sample of each species. Different Latin letters indicate significant differences (two-way ANOVAs with Tukey post hoc tests, *p* < 0.05).

**Figure 4 plants-11-02461-f004:**
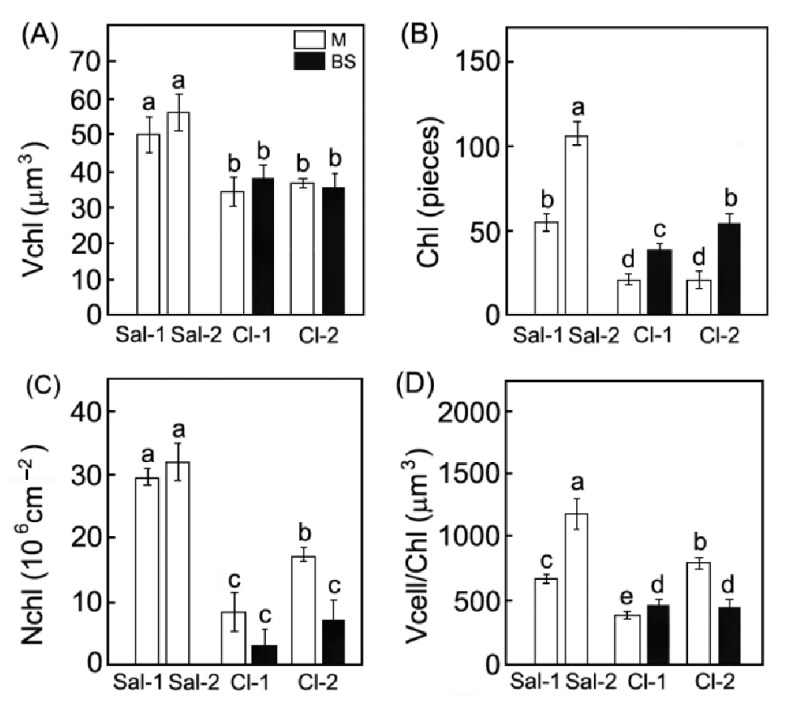
Effect of salinity on the chloroplast volume (**A**) (Vchl), chloroplast number per cell (Chl) (**B**), chloroplast number per unit leaf area (Nchl) (**C**), and cell volume per one chloroplast (Vcell/Chl) (**D**) in plants with different types of photosynthesis: C_3_ *S*. *perennans* (Sal-1, Sal-2) and C_4_ *C*. *crassa* (Cl-1, Cl-2). Data represent means ± SE of *n* = 30 for (**A**–**D**) from cell suspension obtained after the maceration of leaf samples for each species. Different Latin letters indicate significant differences (two-way ANOVAs with Tukey post hoc tests, *p* < 0.05).

**Table 1 plants-11-02461-t001:** Contents of Na^+^, K^+^, and water in the soil of different ecotopes.

Ecotope	Content of Ions, μmol·g^−1^ DW			Soil Water Content, %
	Na^+^	K^+^	Na^+^/K^+^	
Sal-1	105.6 ± 1.8 ^b^	2.3 ± 0.1 ^a^	45.9	33.0 ±3.0 ^a^
Sal-2	165.7 ± 4.8 ^a^	1.9 ± 0.1 ^b^	87.2	23.0 ± 2.0 ^b^
Cl-1	7.4 ± 0.5 ^d^	0.2 ± 0.1 ^d^	37.0	4.0 ± 0.2 ^c^
Cl-2	18.9 ± 5.0 ^c^	0.8 ± 0.1 ^c^	23.6	2.0 ± 0.1 ^d^

Note: Values are the means of three replicates. Different letters indicate significant differences (two-way ANOVAs with Tukey post hoc tests, *p* < 0.05).

**Table 2 plants-11-02461-t002:** Integral parameters of leaf mesostructure in leaves of *S. perennans* and *C. crassa*.

Parameters	*S. perennans* (C_3_)		*C. crassa* (C_4_-NAD-ME)	
	Sal-1	Sal-2	Cl-1	Cl-2
Tleaf	1840 ± 210 ^b^	2300 ± 123 ^a^	870 ± 12 ^d^	1370 ± 142 ^c^
Ncell(M)/Ncell(BS)	–	–	3.4	2.6
Nchl	29.6 ± 1.6 ^a^	32.2 ± 1.2 ^a^	12.4 ± 2.1 ^c^	14.0 ± 4.1 ^b^
Ames/A	36.3	40.5	13.7	24.1
Achl/A	10.8	11.2	4.3	8.7

Note: Data for leaf thickness represent means ± SE of *n* = 20 from leaf cross sections of each species. Different letters indicate significant differences (two-way ANOVAs with Tukey post hoc tests, *p* < 0.05). Abbreviations: Tleaf, the leaf thickness, μm; Ncell(M)/Ncell(BS), the ratio of mesophyll-to-bundle sheath cells; Nchl, the chloroplast number s per unit leaf area; Ames/A, the total surface area of mesophyll and bundle sheath cells per unit leaf area, cm^2^ cm^−2^; Achl/A, is the total surface area of chloroplasts of mesophyll and bundle sheath cells per unit of leaf area, cm^2^ cm^−2^.

**Table 3 plants-11-02461-t003:** Composition of chloroplast lipids (mg·g^−1^ DW) in leaves of *S. perennans* and *C. crassa*.

Lipids	*S. perennans* (C_3_)		*C. crassa* (C_4_)	
	Sal-1	Sal-2	Cl-1	Cl-2
MGDG	1.9 ± 0.1 ^b^ (30.3) *	2.5 ± 0.2 ^a^ (23.9)	1.3 ± 0.1 ^c^ (25.2)	1.3 ± 0.1 ^c^ (26.6)
DGDG	1.3 ± 0.1 ^b^ (21.4)	2.3 ±0.2 ^a^ (21.7)	1.0 ± 0.6 ^b^ (20.3)	1.0 ± 0.05 ^b^ (19.2)
SQDG	0.3 ± 0.02 ^c^ (5.2)	0.9 ± 0.07 ^a^ (8.7)	0.5 ± 0.04 ^b^ (9.3)	0.4 ± 0.03 ^b^ (9.1)
PG	0.4 ± 0.03 ^b^ (6.6)	1.0 ± 0.1 ^a^ (9.0)	0.4 ± 0.03 ^b^ (6.9)	0.4 ± 0.04 ^b^ (7.4)
PC	0.9 ± 0.07 ^b^ (14.2)	2.0 ± 0.1 ^a^ (19.0)	0.8 ± 0.06 ^b^ (15.0)	0.8 ± 0.08 ^b^ (17.3)
PE	0.2 ± 0.02 ^a^ (3.8)	0.2 ± 0.01 ^a^ (1.7)	0.1 ± 0.0 ^b^ (2.7)	0.2 ± 0.02 ^a^ (4.6)
PI	0.2 ± 0.01 ^a^ (2.8)	0.2 ± 0.02 ^a^ (1.7)	0.1 ± 0.08 ^b^ (2.1)	0.1 ± 0.01 ^b^ (1.6)
PA	0.7 ± 0.05 ^a^ (10.9)	0.7 ± 0.06 ^a^ (6.6)	0.6 ± 0.04 ^b^ (11.0)	0.6 ± 0.06 ^b^ (11.9)
PS	0	0	0.2 ± 0.02 (4.6)	0
ST	0.3 ± 0.02 ^a^ (4.5)	0.6 ± 0.05 ^b^ (6.0)	0.1 ± 0.01 ^c^ (2.5)	0.1 ± 0.01 ^e^ (1.7)
Sum	6.2	10.4	5.1	4.9
MGDG/DGDG	1.5	1.1	1.3	1.3
MGDG + DGDG/ SQDG + PG	4.4	2.6	2.8	2. 8
SQDG/PG	0.8	1.0	1.3	1.24
PC/PE	3.79	11.13	5.48	3.74

Note: In brackets is the percentage of the total membrane lipids. Different letters indicate significant differences (two-way ANOVAs with Tukey post hoc tests, *p* < 0.05). * in parentheses shows the content of lipids as a percentage of their total. Abbreviations: MGDG, monogalactosyldiacylglycerol; DGDG, digalactosyldiacylglycerol; SQDG, sulfoquinovosyldiacylglycerol; PG, phosphatidylglycerol; PC, phosphatidylcholine; PE, phosphatidylethanolamine; PI, phosphatidylinositol; PA, phosphatidic acid; ST, sterol.

**Table 4 plants-11-02461-t004:** Fatty acid composition of chloroplast lipids extracted from the halophytes with different type of photosynthesis (% of sum fatty acid).

FA	Species			
	*S. perennans* (C_3_)		*C. crassa* (C_4_)	
	Sal-1	Sal-2	Cl-1	Cl-2
16:0	21.5 ± 0.9 ^c^	26.3 ± 1.1 ^a^	19.9 ± 1.7 ^c^	23.0 ± 0.5 ^b^
18:0	2.1 ± 0.2 ^a^	2.5 ± 0.2 ^a^	1.9 ± 0.2 ^a^	2.4 ± 0.2 ^a^
18:1n9c	2.7 ± 0.3 ^b^	3.3 ± 0.3 ^b^	15.3 ± 1.2 ^a^	16.7 ± 1.4 ^a^
18:2n6c	17.5 ± 1.1 ^a^	17.6 ± 1.5 ^a^	12.4 ± 0.9 ^b^	13.2 ± 1.1 ^b^
18:3n3	48.9 ± 2.2 ^a^	44.5 ± 1.8 ^b^	44.3 ± 1.5 ^b^	37.3 ± 2.5 ^c^
18:2/C18:3	0.36	0.40	0.28	0.35
Others FA	7.3 ± 0.8 ^a^	5.8 ± 0.5 ^b^	6.2 ± 0.6 ^ab^	7.4 ± 0.7 ^a^

Note: Different letters indicate significant differences (two-way ANOVAs with Tukey post hoc tests, *p* < 0.05).

**Table 5 plants-11-02461-t005:** The ratios of Na^+^ and K^+^ in plants and soils.

Ions	*S. perennans* (C_3_)	*C. crassa* (C_4_)
Na^+^_plant_/Na^+^_soil_ Sal-1, Cl-1	67.7 ± 1.9 ^b^	1008 ± 121 ^a^
Na^+^_plant_/Na^+^_soil_ Sal-2, Cl-2	42.5 ± 2.1 ^b^	375 ± 85 ^a^
K^+^_plant_/K^+^_soil_ Sal-1, Cl-1	53.2 ± 1.8 ^b^	469 ± 110 ^a^
K^+^_plant_/K^+^_soil_ Sal-2, Cl-2	84.9 ± 15.0 ^b^	134 ± 14 ^a^
Ratio of Na^+^ content in soils Sal-2/Sal-1 or Cl-2/Cl-1	1.6	2.6
Ratio of Na^+^_plant_/Na^+^_soil_ from Sal-1/Sal-2 or Cl-1/Cl-2	1.6	2.7
Ratio of K^+^ content in soils Sal-1/Sal-2 or Cl-2/Cl-1	1.2	4.0
Ratio of K^+^_plant_/K^+^_soil_ from Sal-1/Sal-2 or Cl-2/Cl-1	1.6	3.5
Net selectivity (net S_K_:_Na_) Sal-1 or Cl-1	0.8	0.5
net S_K_:_Na_ Sal-2 or Cl-2	1.5	0.4

Different letters indicate significant differences (two-way ANOVAs with Tukey post hoc tests, *p* < 0.05).

## Data Availability

All data is original and included in this article.

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
