# Peer review of "Effect of Salinity on Leaf Functional Traits and Chloroplast Lipids Composition in Two C3 and C4 Chenopodiaceae Halophytes"

_plants, 2022, doi:10.3390/plants11192461_

Round 1

Reviewer 1 Report

In my opinion, the manuscript is well written. The ideas are exposed correctly and the comparison between C3 and C4 species response to salinity sound interesting. But in general, the introduction generates expectations that are not fulfill later in the results, discussion and conclusions.

Some specific problems:

The experimental design needs clarification. For instance, in the statistics the authors performed a two-way Anova. However, there is no information about the two analyzed factors, and about their possible interaction  neither in section 4.1 ...conditions of the study nor in section 4.7 statistical analyses.

The discussion section is somehow speculative. Most of information is based on leaf/cell morphological or anatomical differences and contents of some simple molecules (proline) and more complex metabolites (Fatty acids). The article lacks functional evidences that support some of the descriptions. So, for this reason authors are forced to speculate about functional roles. 

Some important questions remain inconclusive and should be discussed: 1. Is there a causal or evolutive connection between salt tolerance traits studied and photosynthetic metabolism C3 or C4? 2. How general are these results? hence how general are the conclusions? Are they applied to any comparison between C3 and C4., Just for these two Chenopodiaceae or only comparing the responses of C3 and C4 (NAD-Me)? So, the limitations of the study should be discussed.

The conclusion section is poor. It is really a summary of the result. I expected some new ideas extracted from the results and linked to the hypothesis. 

other minor problems

1. There is confusion with the concept of ecotope. In Fig 1 seems that ecotope is the sample unit or the replicate. Check this statement "Data represent means ± SE of n = 3 individual ecotopes (lines112-113). From table 1 and Fig 1 it seems that ecotopes and their differences are important for this study, but this is not mentioned in the experimental design, section  4.1 ...conditions of the study.  Is ecotope one of the factors of the two-way anova or the sample unit?

2. Line 333: μM m–2 s–1 Units of PPFD should be reviewed.

Author Response

In my opinion, the manuscript is well written. The ideas are exposed correctly and the comparison between C3 and C4 species response to salinity sound interesting. But in general, the introduction generates expectations that are not fulfill later in the results, discussion and conclusions.

Some specific problems:

Reviewer: The experimental design needs clarification. For instance, in the statistics the authors performed a two-way Anova. However, there is no information about the two analyzed factors, and about their possible interaction  neither in section 4.1 ...conditions of the study nor in section 4.7 statistical analyses.

Author: when analyzing the data, the type of photosynthesis of euhalophytes (two species) and the level of salinity (different ecotopes) were taken into account. In section 4.1 added: Two halophyte species Salicornia perennans Willd. (C3-type of photosynthesis) and Climacoptera crassa (Bied.) Botsch. (C4-NAD-type) were selected for study. Considering that plant responses to natural and artificial growing conditions are significantly different [40], the experimental material was taken from specially selected ecotopes in the basin of the hypersaline lake Elton with different soil salinity.

Reviewer: The discussion section is somehow speculative. Most of information is based on leaf/cell morphological or anatomical differences and contents of some simple molecules (proline) and more complex metabolites (Fatty acids). The article lacks functional evidences that support some of the descriptions. So, for this reason authors are forced to speculate about functional roles.

Author:  agreed, corrected

Reviewer: Some important questions remain inconclusive and should be discussed: 1. Is there a causal or evolutive connection between salt tolerance traits studied and photosynthetic metabolism C3 or C4? 2. How general are these results? hence how general are the conclusions? Are they applied to any comparison between C3 and C4., Just for these two Chenopodiaceae or only comparing the responses of C3 and C4 (NAD-Me)? So, the limitations of the study should be discussed.

Author: There is the direct causal and evolute connection between studied traits of salt tolerance and a type of photosynthetic metabolism. Increase in mesophyll cell volume and succulence are shown as common features of many C3- and C4-species which are tolerant to high salinity (Balnokin et al., 2005; Voronin et al. 2019). Our results really demonstrated 4-fold increase of cell size in C3-species S. perennans and a 1.8-fold increase in the volume of bundle sheath cells in C4-plant C. crassa under salinity. At the same time, we suppose that C3-plants have a larger variation in cell size response to salinity than C4-plants. The occurrence of C4-syndrom is another mechanism of salt tolerance different and independent from succulence. It is shown, that the abundance of C4-dicots correlates with salinity (Pyankov et al., 2000; Feldman et al., 2008; Voronin et al., 2019). Therefore, found differences in cell volume and in the limits of changes in cell sizes between C3 and C4 taxonomically relative annuals demonstrate the different evolute mechanisms of response to salinity in C3 and in C4 plants.

We believe that studied functional traits have different relevance to plant salt adaptation. Some traits are applied to C3 and C4 plants in general. High mesophyll volume as well as larger variation in quantitative leaf traits in C3, especially in mesophyll cell volume, in comparison to C4, belong to the general features of C3-halophytes (Bose et al., 2017; Voronin et al. 2019). Some other traits are characteristic to definite plant functional types - C3-dicots or C4 (NAD-Me)-plants. It is known, that the abundance of C4-dicots correlates with salinity (Feldman et al., 2008; Voronin et al., 2019) whereas C4-monocots do not have such relation. The climatic distribution of C4-monocots is related to air temperature (Teeri, Stowe, 1976; Pyankov, Mokronosov, 1993; Pyankov et al, 2000).  Other types of C3 halophytes – non succulents and other types of C4-plants – C4-monocots will have different traits. Within C3-halophytes, such traits specific for succulent and non-succulent are leaf mass per area, cell volume, cell number, total area of mesophyll cell surface per leaf area (Ivanova et al., 2019). The ratio in quantitative traits between mesophyll and bundle sheath (M/BS) – cell number M/BS, cell volume M/BS, chloroplast number M/BS, chloroplast volume M/BS and the variation in interface area between mesophyll and bundle sheath cells – are specific for different C4-photosynthesis subtypes (Pyankov et al., 2001, 2002). Some studied functional traits are specific for these studied species – leaf thickness and chloroplast number per cell are shown to be strongly species-specific in plants of sub-arid conditions (Yudina et al., 2017).

Reviewer: The conclusion section is poor. It is really a summary of the result. I expected some new ideas extracted from the results and linked to the hypothesis.

Author: In this work, we hypothesized that plants with different types of photosynthetic metabolism can differ significantly in mechanisms and the direction of changes in the structure and function of leaves, cells, chloroplast, as well as photosynthetic membranes with salinity. Thus, the C3 species S. perennans is characterized by a rather variable traits of the photosynthetic apparatus – both at the level of cell morphology and the level of chloroplast membrane systems (envelope and thylakoid membranes). As compared to the C4 species C. crassa, it also has more effective osmoregulatory and protecting mechanisms against the toxic effect of Na+. As a result, S. perennans can withstand a high level of soil salinity subject to high soil humidity. The C4 species C. crassa grows on less saline and more dry soils, i.e., under the conditions of milder ionic but more severe osmotic stress. Adaptation of photosynthetic apparatus in C. crassa to salinity is aimed at the expansion of the intraleaf CO2-diffusion surface, expansion of the interface area between mesophyll and bundle sheath cells due to the increasing the volume of the latter. These adaptations should compensate for scarce CO2 supply when water is in deficit. Overall, the data obtained allow us to conclude that the two Chenopodiaceae species with C3 and C4 types of photosynthetic metabolism developed essentially different mechanisms of adaptation to both at the cellular and subcellular levels.

other minor problems

Reviewer: There is confusion with the concept of ecotope. In Fig 1 seems that ecotope is the sample unit or the replicate. Check this statement "Data represent means ± SE of n = 3 individual ecotopes (lines112-113). From table 1 and Fig 1 it seems that ecotopes and their differences are important for this study, but this is not mentioned in the experimental design, section  4.1 ...conditions of the study.  Is ecotope one of the factors of the two-way anova or the sample unit?

Author:  Data represent means ± SE of n = 3 for each ecotopes. Yes, two halophyte species Salicornia perennans Willd. (C3-type of photosynthesis) and Climacoptera crassa (Bied.) Botsch. (C4-NAD-type) were selected for study. Considering that plant responses to natural and artificial growing conditions are significantly different [40], the experimental material was taken from specially selected ecotopes in the basin of the hypersaline lake Elton with different soil salinity.

Reviewer: Line 333: μM m–2 s–1 Units of PPFD should be reviewed.

Author: corrected – PPFD (1000–2000 µmol m–2 s–1)

Reviewer 2 Report

I have reviewed the manuscript by Rozentsvet et al. submitted to Plants as an article. This paper's authors characterized two C3 and C4 Chenopodiacea halophytes grown in different salinity ecotopes. As the authors claimed, C4 photosynthesis differs from C3 photosynthesis in terms of cell structure, metabolisms, etc. In this manuscript, the authors focused on the cell and chloroplast number, volume, and lipid compositions. However, this reviewer could not understand why these traits are evaluated since other essential characteristics of C3 and C4 photosynthesis may be affected by salinity stresses. 

Importantly, considering the authors took the experimental materials from natural ecotopes, it is not clear whether salinity conditions solely cause the observed traits or not. For instance, the levels of MDA and proline were not correlated with salinity levels of respective ecotopes. Sal-1 grown less salinity condition showed higher accumulation of these compounds.  

From the presented data, it is difficult to evaluate which is cause and effect. As any molecular mechanism of salinity responses is clarified, This reviewer could find the marginal significance of this manuscript. 

Author Response

Reviewer: I have reviewed the manuscript by Rozentsvet et al. submitted to Plants as an article. This paper's authors characterized two C3 and C4 Chenopodiacea halophytes grown in different salinity ecotopes. As the authors claimed, C4 photosynthesis differs from C3 photosynthesis in terms of cell structure, metabolisms, etc. In this manuscript, the authors focused on the cell and chloroplast number, volume, and lipid compositions. However, this reviewer could not understand why these traits are evaluated since other essential characteristics of C3 and C4 photosynthesis may be affected by salinity stresses. 

Author: in this manuscript, the authors focused on the number, volume, and lipid composition of cells and chloroplasts. These signs were evaluated, since one of the key mechanisms of adaptation of halophytes to salinity is the structural and functional rearrangement of the photosynthetic apparatus. It can occur by changing the anatomical structure of the leaf, the quantitative indicators of photosynthetic tissues, the content of photosynthetic pigments, and changing the structure of thylakoid membranes (Ma et al., 2012; Rozentsvet et al., 2018; Kobayaschi, 2016).

Reviewer: importantly, considering the authors took the experimental materials from natural ecotopes, it is not clear whether salinity conditions solely cause the observed traits or not. For instance, the levels of MDA and proline were not correlated with salinity levels of respective ecotopes. Sal-1 grown less salinity condition showed higher accumulation of these compounds. From the presented data, it is difficult to evaluate which is cause and effect. As any molecular mechanism of salinity responses is clarified, This reviewer could find the marginal significance of this manuscript.

Author:  yes, the authors took experimental materials from natural ecotopes. Under these conditions, halophyte plants are in their natural environment and completely complete their life cycle. Therefore, the lack of correlation between MDA (and proline) and salinity levels may be due to other factors. For example, with the content of water in the tissues. Moreover, our results showed (Fig. 2) that MDA and proline correlate well with each other for each plant species.

Reviewer 3 Report

A brief summary

The present manuscript by Rozentsvet et al. describes results of the study about different strategies of C3 and C4 halophyte plants in adaptation to varying salinity levels.

The scientific level of the manuscript, in my opinion, is high and there is no doubt that the manuscript should be published. Nevertheless, I have a few questions, and this makes me choose a «major revision», but I'm sure in the quick response of the authors.

 Broad comments

Lines 72-74. I doubt that the formation and functioning of the chloroplasts, including the grana formation, depend mainly on pigments and lipids compositions. For example, the involvement of bivalent ions into the grana formation is well known. I would recommend to rewrite this sentence more carefully.  

Lines 75-76. In the present view, it seems that pigments of the thylakoid membrane are only sites of light reactions of photosynthesis, but this is not true, because PSII and PSI, for example, are pigment-protein complexes. I would recommend to change  “act as sites” on “involved into”.

Line 98. Can the authors indicate more clearly the location of the region of the study in the beginning of the Results?

The authors do not pay attention in the manuscript to the total carotenoid content in the studied plants, however, this is well known that plants may accumulate carotenoids rapidly in response to abiotic stress as a way to photoprotect their photosynthetic apparatus. In addition, the authors used the method by Lichtenthaler, which allows to calculate the content of carotenoids (absorption at 470 nm) and chlorophylls from the same absorption spectra. Can the authors add these data to the manuscript and discuss them?

Line 414. The authors should specify specific color reactions.

 Specific comments

Line 179. dry weight DW → dry weight (DW)

Line 301. “The pigment content” →This is not correct to talk about pigments without data on the content of carotenoids.

Line 301. Fig 2A →2B

Author Response

Broad comments

Reviewer: Lines 72-74. I doubt that the formation and functioning of the chloroplasts, including the grana formation, depend mainly on pigments and lipids compositions. For example, the involvement of bivalent ions into the grana formation is well known. I would recommend to rewrite this sentence more carefully.

Author: corrected on lipids provide mobility and dynamic properties of chloroplast membranes.

Reviewer: Lines 75-76. In the present view, it seems that pigments of the thylakoid membrane are only sites of light reactions of photosynthesis, but this is not true, because PSII and PSI, for example, are pigment-protein complexes. I would recommend to change “act as sites” on “involved into”.

Author: corrected

Reviewer: Line 98. Can the authors indicate more clearly the location of the region of the study in the beginning of the Results?

Author: added  Russia, Volgograd region, Lake Elton (49°07´N. latitude, 46°50´E. longitude)

Reviewer: The authors do not pay attention in the manuscript to the total carotenoid content in the studied plants, however, this is well known that plants may accumulate carotenoids rapidly in response to abiotic stress as a way to photoprotect their photosynthetic apparatus. In addition, the authors used the method by Lichtenthaler, which allows to calculate the content of carotenoids (absorption at 470 nm) and chlorophylls from the same absorption spectra. Can the authors add these data to the manuscript and discuss them?

Author:  Carotenoids content: Sal-1 0.70 mg g -1 DW, Sal-2 – 0.80 mg g -1 DW, Cl 1- 0.51 mg g -1 DW, Cl-2 0.54 mg g -1 DW. Data were not presented in the text as differences between biotopes were not significant.

Reviewer: Line 414. The authors should specify specific color reactions.

Author: added

Specific comments

Reviewer: Line 179. dry weight DW → dry weight (DW)

Author: corrected

Reviewer: Line 301. “The pigment content” →This is not correct to talk about pigments without data on the content of carotenoids.

Author: corrected

Reviewer:Line 301. Fig 2A →2B

Author: corrected

Reviewer 4 Report

The manuscript compares the effect of salinity in two Salicornia perennans ecotypes and two Climacoptera crassa ecotypes. Authors describe changes in cell morphology, ion content, chlorophyll, proline, MDA and lipids. The experimental approach is correct, although minor details should be completed. Introduction and Discussion sections must be shortened avoiding reiteration of facts well-known in bibliography. The scientific meaning of many expressions is confusing, and the English should be thoroughly revised. I detail bellow specific points that must be addressed.

1. In the title and several places in the text (e.g. the heading of section 2.3), authors claim that they have investigated “functional traits”, which is not correct, “functional” should be changed to “morphological”.

2. Lines 41-42, “the pathway of photosynthesis plays an important ecological role”. The statement is confusing, what is “an ecological role”. Look for a more precise (physiological? adaptative?  ..) role of delete.

3. Lines 62 and 66 change “assimilation” to “fixation”.

4. Line 92. Be concrete: “lake Elton (Rusia)”. As far as I know, there are two Elton lakes one in Russia and other in Canada.

5. Lines 121-123. Only one species is assayed of C3 and C4 plants. Therefore, the generalization of MDA changes to all C3 and C4 plants is only tentative and should be cautiously indicated.

6. Section 4.5. The percentage of chloroplast purity must be indicated. 

Author Response

Reviewer 4

The manuscript compares the effect of salinity in two Salicornia perennans ecotypes and two Climacoptera crassa ecotypes. Authors describe changes in cell morphology, ion content, chlorophyll, proline, MDA and lipids. The experimental approach is correct, although minor details should be completed. Introduction and Discussion sections must be shortened avoiding reiteration of facts well-known in bibliography. The scientific meaning of many expressions is confusing, and the English should be thoroughly revised. I detail bellow specific points that must be addressed.

Author: Agree, and the introduction was shortened. The discussion has been rewritten.

Reviewer: In the title and several places in the text (e.g. the heading of section 2.3), authors claim that they have investigated “functional traits”, which is not correct, “functional” should be changed to “morphological”.

Author: corrected

Reviewer: Lines 41-42, “the pathway of photosynthesis plays an important ecological role”. The statement is confusing, what is “an ecological role”. Look for a more precise (physiological? adaptative?  ..) role of delete.

Author: corrected

Reviewer: Lines 62 and 66 change “assimilation” to “fixation”.

Author: corrected

Reviewer: Line 92. Be concrete: “lake Elton (Rusia)”. As far as I know, there are two Elton lakes one in Russia and other in Canada.

Author: corrected.   Lake Elton, Volgograd region, Russia

Reviewer:  Lines 121-123. Only one species is assayed of C3 and C4 plants. Therefore, the generalization of MDA changes to all C3 and C4 plants is only tentative and should be cautiously indicated.

Author: corrected

Reviewer: Section 4.5. The percentage of chloroplast purity must be indicated. 

Author: The purity of chloroplast fractions was monitored using an Axio observer Z1 inverted biological microscope (fig.). In addition, the analysis of the chloroplast fraction showed a high content (at least 60%) in them of thylakoid lipids - MGDG, DGDG, SQDG, PG.

Figure. Light and Electron microscopy of chloroplasts on the example of S. perennens

Round 2

Reviewer 1 Report

I still have problems to understand the corrections. Because there is not always consistency between the authors responses to my comments and the new version of the Text in  the PDF. Is it possible that the authors have upload a wrong version? I would like to request the authors to make a point by point response indicating clearly page and lines where they have made the correction. It seems that highlighted (green) sections in the new PDF version are not always related to my requested editions.

In the response letter, the authors recognized factors “species” (two levels) and factor “Ecotopes” (two levels). Additionally indicates this has included in Mat and Met section 4.1. But in the PDF still cannot see this inclusion. Please include page and lines you have made this modification. Please, also explicit these factors  in section 4.7 statistical analyses.

In my first round of comments I requested some discussion about the limitations of the study. Probably I did not explain myself clear. The authors are trying to describe different adaptative salinity strategies in two halophyte Chenopodiaceae species, Salicornia perennans Willd. (C3-type of photosynthesis) and Climacoptera crassa (Bied.) Botsch (C4-NAD-type). The author state the hypothesis that “plants with different types of photosynthetic metabolism can differ significantly in mechanisms and the direction of changes in the structure and function of leaves, cells, chloroplast, as well as photosynthetic membranes with salinity. In order to test this hypothesis the authors sample plant material from these two species at two different ecotopes. Unfortunately. Ecotopes Sal 1 and Sal 2 significantly differ in soil salinity compared CL 1 and  CL2 (Table 1). The problem is that C3 species comes only from  Sal ecotopes and C4 species only comes from Cl ecotopes. So, my question to the authors is to discuss the limitations of this sampling design to test the hypothesis. Why? Because I think it is difficult to assess how much of the different adaptation strategies found between species  is really due to the contrasting C3 C4 photosynthesis and how much of it is due only to the huge difference in soil salinity observed between the habitat where the species where collected. In my opinion the only way to separate these two effects is a random design where you can put both species at the same contrasting salinities. So please include this possible limitation in you discussion.

Minor problems: I still can see these minor problems in the last version

Reviewer: Line 333 First version: μM m–2 s–1 Units of PPFD should be reviewed. Author: corrected – PPFD (1000–2000 µmol m–2 s–1) this looks correct in the response comments, but in the new  version (Page 9, line 39) PPFD m-2S-1 what happened with the µmol?

Additionally, I have noticed there is a weird combination of Molar  M (mol/L of solution) which is a concentration,  and a content expressed by gram of DW. For instance:

1.     in Table 1 units µM g-1DW. Is this correct? or should be µmol g-1 DW.

2.     Fig 1, Na+ and K+  mM g-1DW. Is this correct? or should be mmol g-1 DW.

3.     Fig 2 MDA µM g-1DW. Is this correct? or should be µmol g-1 DW.

Table 3 has a weird unit g-3g-1DW (% of total). Better use mg g-1 DW as you state in section 2.4 line 12. I am not sure why you mentioned % of total if you have mg g-1 DW This is an absolute measurement. Isn’t it?

Table 4: Fatty acid compositions does not have units. Please include it.

Author Response

Dear Reviewer,

  Thank you very much for your careful reading of our work. The authors have made all the corrections to the article.

Reviewer:  the PDF. Is it possible that the authors have upload a wrong version? I would like to request the authors to make a point by point response indicating clearly page and lines where they have made the correction. It seems that highlighted (green) sections in the new PDF version are not always related to my requested editions.

In the response letter, the authors recognized factors “species” (two levels) and factor “Ecotopes” (two levels). Additionally indicates this has included in Mat and Met section 4.1. But in the PDF still cannot see this inclusion. Please include page and lines you have made this modification. Please, also explicit these factors  in section 4.7 statistical analyses.

Author: This part was contributed to 2. Results

2.1. Contents of Na+ and K+ in the leaves (Page 3, line 12-22)  and in section 4.7 statistical analyses (page 14, line 35-36).

In my first round of comments I requested some discussion about the limitations of the study. Probably I did not explain myself clear. The authors are trying to describe different adaptative salinity strategies in two halophyte Chenopodiaceae species, Salicornia perennans Willd. (C3-type of photosynthesis) and Climacoptera crassa (Bied.) Botsch (C4-NAD-type). The author state the hypothesis that “plants with different types of photosynthetic metabolism can differ significantly in mechanisms and the direction of changes in the structure and function of leaves, cells, chloroplast, as well as photosynthetic membranes with salinity. In order to test this hypothesis the authors sample plant material from these two species at two different ecotopes. Unfortunately. Ecotopes Sal 1 and Sal 2 significantly differ in soil salinity compared CL 1 and  CL2 (Table 1). The problem is that C3 species comes only from  Sal ecotopes and C4 species only comes from Cl ecotopes. So, my question to the authors is to discuss the limitations of this sampling design to test the hypothesis. Why? Because I think it is difficult to assess how much of the different adaptation strategies found between species  is really due to the contrasting C3 C4 photosynthesis and how much of it is due only to the huge difference in soil salinity observed between the habitat where the species where collected. In my opinion the only way to separate these two effects is a random design where you can put both species at the same contrasting salinities. So please include this possible limitation in you discussion.

Author:  Dear reviewer, you are absolutely right and it would be fair to compare C3 and C4 plants in ecotopes with equal salinity levels. However, under natural conditions, all halophytes occupy strictly defined ecotopes with a certain salinity and soil moisture (Sukhorukov, 2014, Rozentsvet et al., 2020(Handbook of Halophytes  From Molecules to Ecosystems towards Biosaline Agriculture» «Lipids of Halophyte Species Growing in Lake Elton Region (South East of the European Part of Russia)»  Olga A. Rozentsvet, Viktor N. Nesterov, Elena S. Bogdanova https://doi.org/10.1007/978-3-030-17854-3_114-1)). We deliberately searched for ecotopes in which each species grew in contrasting conditions. Therefore, we believe that such an approach is fully justified.

Minor problems: I still can see these minor problems in the last version

Reviewer: Line 333 First version: μM m–2 s–1 Units of PPFD should be reviewed. Author: corrected – PPFD (1000–2000 µmol m–2 s–1) this looks correct in the response comments, but in the new  version (Page 9, line 39) PPFD m-2S-1 what happened with the µmol?

Additionally, I have noticed there is a weird combination of Molar  M (mol/L of solution) which is a concentration,  and a content expressed by gram of DW. For instance:

Author: corrected (Page 12, line 10)

Reviewer:  1.     in Table 1 units µM g-1DW. Is this correct? or should be µmol g-1 DW.

Author: corrected (Page 7, line 12)

Reviewer:  2.     Fig 1, Na+ and K+  mM g-1DW. Is this correct? or should be mmol g-1 DW.

Author: corrected (Page 5, line 11)

Reviewer:  3.     Fig 2 MDA µM g-1DW. Is this correct? or should be µmol g-1 DW.

Author: corrected (Page 6, line 1)

Reviewer:  Table 3 has a weird unit g-3g-1DW (% of total). Better use mg g-1 DW as you state in section 2.4 line 12. I am not sure why you mentioned % of total if you have mg g-1 DW This is an absolute measurement. Isn’t it?

Author: corrected  (*   in parentheses shows the content of lipids as a percentage of their total)(page 8, line 11, 14).

Reviewer:  Table 4: Fatty acid compositions does not have units. Please include it.

Author: corrected  (% of sum fatty acids)(page 9, line 7).

Reviewer 3 Report

As I have already said, there is no doubt about the high scientific level of the manuscript. In addition, the authors gave exhaustive answers to all my comments after the first round. Thus, I think that the manuscript of Rosencvet et al. can be published in its current form.

Author Response

The authors would like to thank you for evaluating our manuscript. We have tried to properly resolve all doubts, and we believe that our article has improved significantly.

Reviewer 4 Report

Th last version of the manuscript properly addressed the main questions raised to the first version.

Author Response

Dear Reviewer,

 We are grateful for valuable comments,  which have helped us to substantially improve the manuscript.

With best wishes,

Ph.  Elena Bogdanova.